# An Embedding is Worth a Thousand Noisy Labels

**Francesco Di Salvo**                                   *francesco.di-salvo@uni-bamberg.de*
*xAILab Bamberg*
*University of Bamberg, Germany*

**Sebastian Doerrich**                                   *sebastian.doerrich@uni-bamberg.de*
*xAILab Bamberg*
*University of Bamberg, Germany*

**Ines Rieger**                                          *ines.rieger@uni-bamberg.de*
*xAILab Bamberg*
*University of Bamberg, Germany*

**Christian Ledig**                                      *christian.ledig@uni-bamberg.de*
*xAILab Bamberg*
*University of Bamberg, Germany*

**Reviewed on OpenReview:** *https: // openreview. net/ forum? id= X3gSvQjShh*

## Abstract

The performance of deep neural networks scales with dataset size and label quality, rendering the efficient mitigation of low-quality data annotations crucial for building robust and cost-effective systems. Existing strategies to address label noise exhibit severe limitations due to computational complexity and application dependency. In this work, we propose `WANN`, a Weighted Adaptive Nearest Neighbor approach that builds on self-supervised feature representations obtained from foundation models. To guide the weighted voting scheme, we introduce a reliability score $\eta$, which measures the likelihood of a data label being correct. `WANN` outperforms reference methods, including a linear layer trained with robust loss functions, on diverse datasets of varying size and under various noise types and severities. `WANN` also exhibits superior generalization on imbalanced data compared to both Adaptive-NNs (`ANN`) and fixed $k$-NNs. Furthermore, the proposed weighting scheme enhances supervised dimensionality reduction under noisy labels. This yields a significant boost in classification performance with $10\times$ and $100\times$ smaller image embeddings, minimizing latency and storage requirements. Our approach, emphasizing efficiency and explainability, emerges as a simple, robust solution to overcome inherent limitations of deep neural network training. The code is available at github.com/francescodisalvo05/wann-noisy-labels.

## 1 Introduction

The remarkable results achieved by deep neural networks in a multitude of domains are possible due to ever-increasing computational power. This progress has enabled the development of deeper architectures with stronger learning capabilities. However, these high-parametric architectures are often characterized as data-hungry, because they require a large amount of data to generalize effectively. Collecting and annotating such extensive datasets can be both expensive and time-consuming, potentially introducing machine and human errors. In fact, the proportion of corrupted labels in real-world datasets ranges from 8% to 38.5% (Song et al., 2022). While unsupervised and semi-supervised methods have garnered significant attention within the research community (Chen et al., 2020; Tarvainen & Valpola, 2017; Berthelot et al., 2019), supervised methods are still widely employed due to their generally higher performance. In this context, incorrect labels pose significant challenges, including a tendency to memorize label noise, negatively affecting generalization

capabilities. This issue, as demonstrated by Zhang et al. (2021), is hindering the adoption of AI systems in safety-critical domains, such as healthcare. Consequently, there is a growing interest in enhancing the robustness of deep models against noisy labels, resulting in five different lines of research (Song et al., 2022): *robust architectures*, *robust regularization*, *robust loss function*, *loss adjustment* and *sample selection*. Nevertheless, as also highlighted by Zhu et al. (2022), all these approaches involve a learning process and, by definition, have limitations in generalizing to diverse datasets or varying noise rates. We address those limitations and the challenges posed due to noisy labels by exploiting feature representations obtained from large pre-trained models, commonly referred to as *foundation models*. Many of these foundation models are now publicly available and designed for a wide range of applications, including image classification, semantic segmentation tasks, and beyond. While their initial focus was primarily on text (Devlin et al., 2019; Brown et al., 2020) and natural images (Dosovitskiy et al., 2021; Caron et al., 2021; Oquab et al., 2023; Radford et al., 2021), there is currently a focus on the development of open-source foundation models tailored to specific domains, such as healthcare (Tu et al., 2024; Li et al., 2023; Zhou et al., 2023; Xu et al., 2024), making it possible to translate this paradigm to various applications with little effort. Specifically, large image-based foundation models are usually trained in a self-supervised fashion, *e.g.*, by means of contrastive learning (Chen et al., 2020), representing similar objects closely within the embedding space, and semantically distinct objects further away. Thus, the position of an image in the embedding space can be just as informative as the label, if not more so. Although embedding-space approaches are not new in the literature to process noisy labels, they typically focus on noise detection, through online (Bahri et al., 2020) or offline (Zhu et al., 2022) methods. However, thanks to the representational power of modern embedding spaces (Radford et al., 2021; Oquab et al., 2023), we believe that it is also possible to provide robust predictions by building on simple $k$-NN methods. This category of approaches not only offers computational efficiency but also enhances explainability. Moreover, the lower reliance on hyperparameters further enhances generalizability. Thus, owing to its simplicity and efficiency, it is able to mitigate some of the limitations associated with the training of deep networks. In this work, we propose a Weighted Adaptive Nearest Neighbor (`WANN`) approach, illustrated in Figure 1, which operates on the image embeddings extracted from a pre-trained foundation model. The introduced weighted adaptive voting scheme addresses the challenges posed by large-scale, limited, and imbalanced noisy datasets. Our contributions are:

- Formulation of a *reliability score* ($\eta$), measuring the likelihood of a label being correct. This scoring guides the formulation of `WANN`, a method that determines a neighborhood of the test sample to weight an adaptively determined number of training samples in the embedding space.

- Extensive quantitative experiments, confirming that `WANN` has overall greater robustness compared to reference methods (`ANN`, fixed $k$-NN, robust loss functions) across diverse datasets and noise levels, including limited and severely imbalanced noisy data scenarios.

- Formulation of the Filtered LDA (`FLDA`) approach that relies on our sample-specific reliability scores for higher quality projections. `FLDA` improves the robustness for dimensionality reduction by filtering out detected noisy samples and improves the classification performances with $10\times$ and $100\times$ smaller image embeddings.

Consequently, through lightweight embeddings and a simple yet efficient classification algorithm, we demonstrate that working efficiently within the embedding space might constitute a potential paradigm shift, increasing efficiency and robustness while addressing critical limitations related to model training.

## 2 Related works

**Noise robust learning** Robust learning from noisy labels has received considerable attention in recent years, and multiple deep-learning-based methods are currently available (Song et al., 2022). Two popular categories of approaches are *sample selection* and *loss adjustment*. Methods based on sample selection aim to isolate correctly labelled samples from the noisy dataset, often employing strategies such as multi-network learning, iterative data filtering, or both (Han et al., 2018; Jiang et al., 2018; Song et al., 2019; Li et al., 2020; Wei et al., 2020; Xiao et al., 2023; Zhang et al., 2024; Fooladgar et al., 2024). In contrast, loss

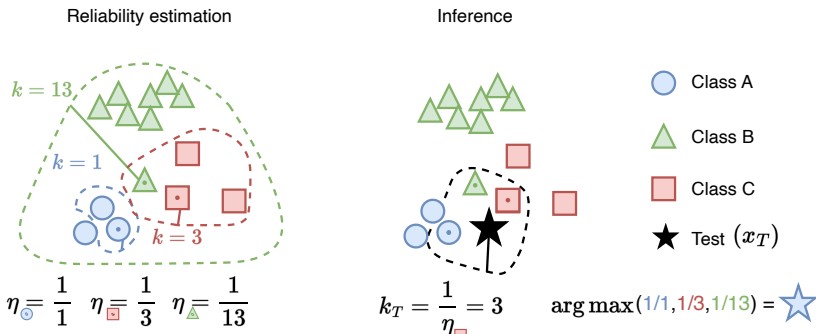

Figure 1: Illustration of the proposed Weighted Adaptive Nearest Neighbor (`WANN`) algorithm. Initially, a *reliability score* ($\eta$) is computed for each training observation, representing the inverse of the minimum number of samples needed for a correct prediction. During inference, the adaptive neighborhood size ($k_T$) of each **test observation** ($x_T$) is determined based on the reliability score of its closest training sample ($k_T = \frac{1}{\eta} = 3$). A weighted majority vote determines the final label, reducing the impact of noisy labels.

adjustment techniques modify how the loss is computed for every sample, thereby mitigating the impact of label noise (Patrini et al., 2017; Tanaka et al., 2018; Liu et al., 2020; Zheng et al., 2021). However, due to their computational complexity, these models often lack generalizability across datasets or noise settings. Some of the lightweight techniques designed to handle noisy labels fall under the category of *robust loss functions*. While the Cross Entropy (CE) suffers from overfitting to wrong labels (Zhang et al., 2021), the Mean Absolute Error (MAE) has been theoretically guaranteed to be noise-tolerant (Ghosh et al., 2017). However, it suffers from severe underfitting in challenging domains. To address this issue and enhance its generalization, the Generalized Cross Entropy (GCE) (Zhang & Sabuncu, 2018) was introduced as a generalization of both MAE and CE. While Wang et al. (2019) propose the Symmetric Cross Entropy (SCE), Zhou et al. (2021) overcome the symmetric condition through Asymmetric Loss Functions. Active Passive Losses (APL) (Ma et al., 2020) which combines two robust losses to balance between overfitting and underfitting. Recently, Active Negative Losses (ANL) (Ye et al., 2023; 2024) replaced the previous *passive loss* (in APL) with a Normalized Negative Loss Function (NNLF). Nevertheless, despite their relatively lower computational complexity compared to multi-network methods, they still inherit challenges related to training neural networks. Some of these challenges include the demand for large datasets, lack of explainability, computational complexity, hyperparameter dependence, and overfitting towards wrong labels.

**$k$-NN for label noise** Besides native deep learning-based approaches, certain traditional machine learning methods are robust by design. Notably, $k$-Nearest-Neighbors-based methods have recently gained attention for their utility in filtering out noisy training observations (Reeve & Kabán, 2019; Bahri et al., 2020; Kong et al., 2020). Although the proposed online $k$-NN approaches can enhance the robustness of DNNs trained with label noise, they do not address the fundamental issues associated with DNNs, such as the lack of explainability, data-efficiency, and generalizability. A closely related work (Zhu et al., 2022) already emphasized the potential of a training-free cleaning strategy based on $k$-NN with CLIP (Radford et al., 2021) embeddings, enhancing DNNs downstream performance. Furthermore, a concurrent work (Zhu et al., 2024) demonstrated that a simple noise-agnostic linear probing on high-quality features outperforms a deep network trained from scratch, achieving state-of-the-art results. In contrast, we show that a training-free approach can outperform linear probing, with or without robust loss functions. Indeed, our work represents a broader paradigm shift: rather than training large networks from scratch, we utilize high-quality feature representations from foundation models and employ a simple, interpretable, and training-free classification scheme in the embedding space. This design choice directly addresses practical challenges such as mitigating overfitting to noisy labels, enabling transparent predictions (*cf.* Figure 2), and simplifying dataset updates. We further enhance classification performance under label noise and label imbalance through the adoption of an adaptive neighborhood for each test observation.

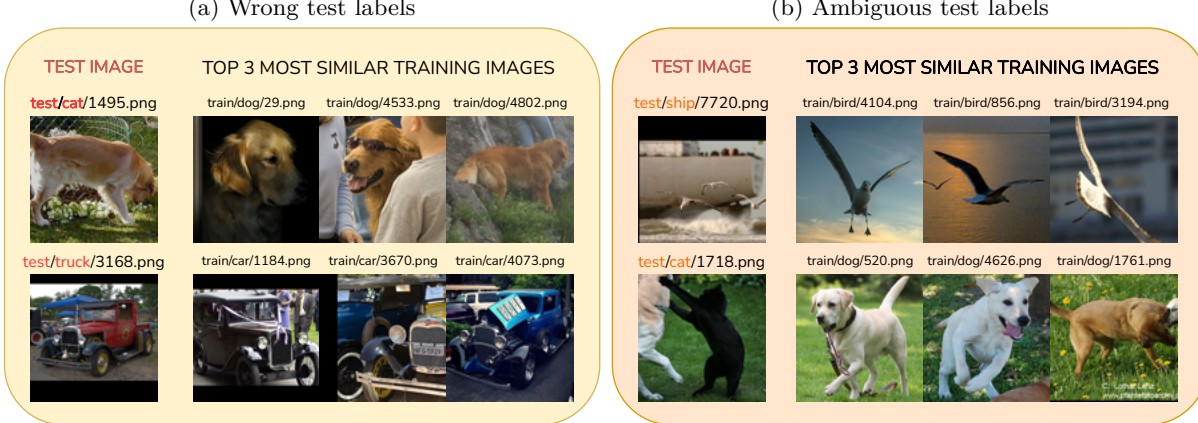

Figure 2: Example of test images and their relative top-3 closest training samples extracted from the STL-10 dataset (Coates et al., 2011). Figure 2a shows two test images having wrong labels, while Figure 2b shows ambiguous labels, including multiple known objects within a single image. For instance, a `bird` with a `ship` in the background, or a `dog` fighting with a `cat`.

## 3  Method

At its core, the proposed method relies on an adaptive $k$-NN search over the image embeddings extracted from a pre-trained foundation model, using the Euclidean distance as reference distance metric. In particular, we employ the DINOv2 (Oquab et al., 2023) Large backbone, with 14×14 patches, yielding image embeddings of size 1024. DINOv2 was trained in a self-supervised fashion on 142M images, and the choice of the backbone is further motivated in Section 4.1. To enhance computational efficiency, we pre-generated a database of embeddings for all of the evaluated datasets, following the approach adopted by Nakata et al. (2022).

**Label reliability score**  Although $k$-NN is robust against noisy labels by design, its efficacy may diminish under severe noise conditions. To address this challenge, the introduction of a weighting scheme becomes helpful, aiming to mitigate the impact of noisy labels. Intuitively, a proper weighting scheme should assign higher weights to samples that are more likely to have a correct label, and lower weights to those potentially mislabeled. In this context, we introduce the concept of *label reliability*, building upon the work of Sun & Huang (2010). Extending their approach, which inferred an optimal $k$ value for each training observation, we determine the reliability (*i.e.,* quality) of the associated label, guiding a subsequent weighting scheme. For a given training observation $(x_i, y_i)$, we define the *reliability* $\eta_i$ as the inverse of the minimum number of training samples (neighborhood size) required for a correct $k$-NN classification. Dropping the index $i$ for readability, we express $\eta$ as the function $\mathcal{H}(x, y; \mathcal{X})$, where $\mathcal{X}$ represents the training dataset:

$$\mathcal{H}(x, y; \mathcal{X}) = \frac{1}{\underset{k' \in [k_{\min}, k_{\max}]}{\arg \min} \; \texttt{kNN}_{\mathcal{X}}(x, k') = y} \tag{1}$$

Here $k_{\min}$ and $k_{\max}$ define the search interval for $k' \in \mathbb{N}$. It was previously shown by Zhu et al. (2022) that for a fixed $k$-NN approach a suitable neighborhood size to detect corrupted labels is 10. Consequently, we set $k_{\min} = 11$ and $k_{\max} = 51$. This allows to establish a substantial search space covering a wide range of noise levels while ensuring a baseline ability to detect corrupted labels ($k_{\min} > 10$). During parameter search we increase $k'$ by 2 at each incremental step, ensuring an odd number of samples in the neighborhood while reducing the computational complexity. If there is no $k'$ satisfying Equation 1, we set the optimal $k = \frac{1}{k_{\max}}$. The correlation between the value of $k$ and the reliability of the assigned label stems from the main property of the embedding space. The proximity of similar objects suggests that a small $k$ should be sufficient for accurate classification. In such cases, the reliability of the data point is higher ($k \downarrow \; \eta \uparrow$).

Conversely, if a data point requires a large $k$ value for correct classification, approaching or reaching $k_{\max}$, it may signal potential mislabeling ($k \uparrow \eta \downarrow$). The pseudocode is presented in Algorithm 1, and it requires three parameters: $k_{\min}, k_{\max}$, and $D_{\text{train}}$. The latter, $D_{\text{train}}$, represents a data structure containing a triple (`key`, $x, y$) for each train observation. Here, `key` serves as a unique identifier.

---

**Algorithm 1** reliability($k_{\min}, k_{\max}, D_{\text{train}}$)

---

$R \leftarrow \{\ \}$      ▷ reliability map of `key : value` pairs
$P \leftarrow \texttt{PairwiseDist}(D_{\text{train}}, D_{\text{train}})$      ▷ matrix
**for** (`key`, $x, y$) **in** $D_{\text{train}}$ **do**
    **for** $k' = k_{\min}, k_{\min} + 2, \ldots, k_{\max}$ **do**
        $\hat{y} \leftarrow \texttt{kNN}_P(x, k')$      ▷ $k$ points excluding $x$
        **if** $\hat{y} == y$ **then**
            $R[\texttt{key}] \leftarrow \frac{1}{k'}$
            break
        **end if**
    **end for**
    **if not** `key` **in** `R.keys()` **then**
        $R[\texttt{key}] \leftarrow \frac{1}{k_{\max}}$
    **end if**
**end for**
**return** $R$

---

**Weighted Adaptive Nearest Neighbor (WANN)**   Adaptive Nearest Neighbors algorithms were commonly employed for tabular data with a limited number of observations and classes (Geva & Sitte, 1991; Wettschereck & Dietterich, 1993; Wang et al., 2006). Following the intuition of Sun & Huang (2010), we use $\eta$ to adaptively determine the neighborhood size for any test observation. In fact, if a test sample is close to a potentially noisy label ($\eta \downarrow$) a larger neighborhood is preferable to reduce variance. Conversely, if a test observation is close to a possibly clean label ($\eta \uparrow$), a small neighborhood increases specificity. Thus, given a test observation $x_T$ with *nearest training sample* $(x_n, y_n, \eta_n)$ (nearest in the embedding space), its test neighborhood size $k_T$ will be adaptively defined by means of the reliability score of $x_n$ (*cf.* Equation 2).

$$k_T = \frac{1}{\eta_n} = k_n \tag{2}$$

With this notation we formulate an Adaptive Nearest Neighbor (`ANN`) method as baseline, similar to Sun & Huang (2010):

$$\texttt{ANN}(x_T) = \arg\max_{c \in C} \frac{1}{k_T} \sum_{i \in N_T} \mathbb{1}(y_i = c) \tag{3}$$

Here, $C$ denotes the set of classes, $N_T$ refers to the adaptive neighborhood determined by the $k_T$ closest observations, and $\mathbb{1}(\cdot)$ is the indicator function. Finally, having defined the *reliability score* as $\eta = \mathcal{H}(x, y; \mathcal{X})$, we enhance `ANN` by introducing a weighting scheme. Thus, instead of merely taking the most frequent class in the neighborhood, we now associate a weight (*i.e.*, $\eta$) to each training observation. Consequently, training samples with higher *reliability* $\eta$ will have a greater impact on the final prediction, while those with lower $\eta$ (possibly wrong labels) will contribute less. As such, our Weighted Adaptive Nearest Neighbor approach is formulated as:

$$\texttt{WANN}(x_T) = \arg\max_{c \in C} \sum_{i \in N_T} \eta_i \mathbb{1}(y_i = c) \tag{4}$$

**Filtered dimensionality reduction**   $k$-NN, like several other distance-based machine learning methods, is susceptible to the *curse of dimensionality* (Kouiroukidis & Evangelidis, 2011). In high-dimensional spaces, the relative distances between data points become less discriminative, inevitably affecting the algorithm's performance. To address this issue, dimensionality reduction techniques can be applied to reduce the dimensionality of the feature space with minimal information loss. Common choices for dimensionality reduction include linear methods like Principal Component Analysis (PCA) (Jolliffe, 2002) and Linear Discriminant Analysis (LDA) (Belhumeur et al., 1996).

While PCA seeks directions of maximum variance, LDA leverages class labels to maximize the separation between classes and minimize variance within classes, according to the Fisher-Rao criterion. Due to its supervised nature, LDA relies on correct labels, as label noise can result in inaccurate linear mappings. To address this limitation, we employ `WANN`'s *reliability score $\eta$* to filter out potentially wrong labels (*i.e.*, $\eta = 1/k_{max}$) before performing LDA. We term this approach Filtered LDA (`FLDA`). Once the projection axes have been obtained through the *clean* dataset, we subsequently project both training and test datasets onto those axes.

## 4   Experiments and results

Initially, in Section 4.1, we justify the choice of the DINOv2 Large backbone (Oquab et al., 2023). Next, in Sections $4.2 - 4.5$, we evaluate `WANN` against robust loss functions on *noisy* real-world, limited, and medical data. In Section 4.6, we assess the robustness of the weighted adaptive neighborhood in the context of heavily imbalanced datasets. We further demonstrate the effectiveness of the proposed Filtered LDA (`FLDA`) approach in Section 4.7. Moreover, in Section 4.8, we qualitatively explore the explainability benefits of our approach on noisy datasets. It is important to note that none of the datasets used in our quantitative experiments were part of DINOv2's pretraining data, as detailed in Table 15 of (Oquab et al., 2023). Finally, Appendix A–B demonstrate the generalizability of `WANN` utilizing an additional backbone and its robustness with respect to the proposed reliability score, respectively.

### 4.1   Backbone

In order to achieve satisfactory $k$-NN performance, a high-quality feature representation is essential to ensure close proximity of similar objects. To evaluate this, we initially compare the performance of ResNet50 and ResNet101 (He et al., 2016), both pre-trained on ImageNet-1k. In the realm of Vision Transformers, we explore self-supervised pre-training with MAE (He et al., 2022) (Base and Large), CLIP pre-training Radford et al. (2021) (Base and Large), and lastly DINOv2 pre-training (Oquab et al., 2023) (Base and Large). All pre-trained models are obtained from HuggingFace (timm). We report `WANN`'s classification accuracy on CIFAR-10 and CIFAR-100 (Krizhevsky et al., 2009), utilizing $(k_{\min}, k_{\max}) = (11, 51)$.

**Results**   Table 1 displays the results, clearly confirming DINOv2 as backbone choice. Results further reveal a notable decline in performance for all backbones as the number of classes increases. This trend is expected due to increasing overlap of class-specific feature distributions within the embedding space. Additionally, obtaining ground truth labels for a larger set of visually more ambiguous classes poses more challenges associated with label quality. Notably, comparing MAE ViTs with ResNets, both pre-trained on ImageNet, it is possible to observe a substantially lower performance of MAE ViTs. Despite the higher number of parameters, the self-supervised pre-training paradigm appears to be not beneficial in this context. Furthermore, the discrepancy between CLIP and DINOv2 can be attributed to the divergent pre-training strategies: CLIP is trained on pairs of image-text, whereas DINOv2 exclusively relies on image data. Beyond computational efficiency, the DINOv2 framework allows to disregard potential ambiguities introduced by textual descriptions. For this reason, it also required considerably less data, as DINOv2 was trained on 142 millions of images while CLIP on 400 millions of image-text pairs. Thus, in scenarios where text embeddings are not explicitly needed, as in our case, opting for an image-centric training paradigm is beneficial.

| Backbone | #Par | #Dim | CIFAR-10 | CIFAR-100 |
|---|---|---|---|---|
| ResNet50 | 26M | 2048 | 84.09 | 60.31 |
| ResNet101 | 45M | 2048 | 87.32 | 64.59 |
| MAE ViT-B/16 | 86M | 768 | 72.04 | 45.18 |
| MAE ViT-L/16 | 303M | 1024 | 82.99 | 56.18 |
| CLIP ViT-B/16 | 86M | 768 | 92.24 | 68.96 |
| CLIP ViT-L/14 | 304M | 1024 | 95.65 | 75.69 |
| DINOv2 ViT-B/14 | 87M | 768 | 98.70 | 89.36 |
| DINOv2 ViT-L/14 | 304M | 1024 | **99.36** | **91.68** |

Table 1: `WANN`'s classification performance (accuracy ↑) over eight different feature spaces, on CIFAR-10 and CIFAR-100, which contains 10 and 100 classes, respectively. Note that "#Par" denotes the number of parameters and "#Dim" denotes the feature dimension.

## 4.2 Real-world noisy labels

We compared `ANN` and `WANN` with lightweight techniques falling into the category of robust loss functions. As commonly employed for assessing the feature space quality (Oquab et al., 2023), we trained a single linear layer over the image embeddings. The embeddings were normalized using training set statistics, excluding 15% for validation in linear training. The linear probing utilizes the Adam optimizer with a learning rate of $1 \times 10^{-4}$ for 100 epochs, with early stopping after 5 epochs. These settings were consistently applied to all experiments in this manuscript. Following the publicly available losses and hyperparameters proposed by Ye et al. (2023), we compared: Cross Entropy (CE) as a baseline, Focal Loss (FL), Mean Absolute Error (MAE), Generalized Cross Entropy (GCE), Symmetric Cross Entropy (SCE), Active Passive Losses (NCE-MAE, NCE-RCE, NFL-RCE), Asymmetric Losses (NCE-AGCE, NCE-AUL, NCE-AEL), and Active Negative Losses (ANL-FL,ANL-CE-ER). Additionally, we include Early Learning Regularization (ELR) (Liu et al., 2020), a widely adopted baseline related to label correction. Since the embeddings are pre-generated, no data augmentation was applied. Both `ANN` and `WANN` use $(k_{\min}, k_{\max}) = (11, 51)$. We measure the accuracy on two publicly available datasets, namely CIFAR-10N and CIFAR-100N (Wei et al., 2022), which are "noisy versions" of the established CIFAR datasets. CIFAR-10N contains three human annotations per image, proposing different noisy training splits with varying levels of noise rates (NR). Conversely, CIFAR-100N presents solely one annotation per image, thereby offering one single noisy split. Due to the potential sensitivity of linear training to the choice of the random seed, we show the accuracy as the mean and standard deviation computed over five runs.

**Results**  Table 2 shows the results of the designed experiment for real-world noisy labels. The introduced weighting scheme in `WANN` substantially enhances robustness compared to `ANN`, particularly in high-noise conditions. Additionally, `WANN` exhibits greater advantages compared to robust loss functions in scenarios with limited noise ratios ($< 20\%$), showing a notable lower performance drop than robust loss functions on CIFAR-10N *Aggr* and *R1*, respectively. Active Negative Losses (ANL-CE-ER and ANL-FE) rank among the top robust loss functions, approaching `WANN`'s performance. Furthermore, while NCE-AGCE exhibits a higher accuracy on CIFAR-100N compared to `WANN`, it notably presents a higher performance drop with respect to the clean data (7.15% versus 6.24%), indicating lower robustness.

## 4.3 Limited noisy data

The inherent limitation of deep models lies in their reliance on a large amount of clean annotated data. In contrast, $k$-NN models demand less data, exhibiting potential robustness, particularly in the presence of label noise. To demonstrate this, two subsets are randomly sampled from CIFAR-10, each containing only 50 and 100 samples per class. This experiment compares robust loss functions against `ANN` and `WANN`.

| Noisy split | CIFAR-10N | | | | | | CIFAR-100N | |
| | Clean | Aggr. | R1 | R2 | R3 | Worst | Clean | Noisy |
| NR ($\approx$) | - | 9.01% | 17.23% | 18.12% | 17.64% | 40.21% | - | 40.20% |
|---|---|---|---|---|---|---|---|---|
| CE | 99.40±0.03 | 98.46±0.07 | 98.29±0.11 | 98.33±0.11 | 98.45±0.05 | 93.20±0.32 | 93.38±0.13 | 83.10±0.29 |
| FL | 99.39±0.02 | 98.36±0.08 | 98.12±0.12 | 98.21±0.07 | 98.20±0.13 | 92.85±0.18 | 93.30±0.10 | 82.03±0.32 |
| MAE | 99.42±0.02 | 99.21±0.02 | **99.22±0.06** | 99.17±0.03 | 99.20±0.06 | 95.79±0.41 | 93.40±0.10 | 86.08±0.25 |
| GCE | 99.39±0.05 | 99.11±0.01 | 99.06±0.03 | 98.96±0.08 | 99.03±0.05 | 95.19±0.43 | 93.45±0.07 | 85.74±0.32 |
| SCE | 99.42±0.02 | 99.14±0.03 | 99.11±0.06 | 99.07±0.08 | 99.12±0.05 | 95.52±0.53 | 93.40±0.14 | 83.15±0.33 |
| ELR | 99.44±0.02 | 99.13±0.10 | 99.09±0.08 | 99.14±0.07 | 99.14±0.04 | 97.72±0.37 | 93.34±0.04 | **86.10±0.22** |
| NCE-MAE | 99.42±0.01 | 99.17±0.02 | 99.15±0.04 | 99.17±0.02 | 99.15±0.05 | 95.35±0.35 | 93.46±0.08 | 85.44±0.40 |
| NCE-RCE | 99.42±0.02 | 99.21±0.03 | 99.21±0.07 | 99.15±0.04 | 99.19±0.08 | 95.58±0.35 | 93.47±0.08 | 86.08±0.08 |
| NFL-RCE | 99.42±0.02 | 99.21±0.03 | 99.21±0.07 | 99.15±0.04 | 99.19±0.08 | 95.58±0.35 | 93.46±0.07 | 86.05±0.11 |
| NCE-AGCE | 99.42±0.02 | 99.21±0.03 | 99.21±0.05 | 99.18±0.03 | 99.18±0.06 | 95.93±0.43 | 93.40±0.07 | **86.25±0.12** |
| NCE-AUL | 99.42±0.02 | 99.20±0.03 | 99.21±0.06 | 99.18±0.04 | 99.18±0.05 | 95.77±0.48 | 93.41±0.09 | 85.81±0.27 |
| NCE-AEL | 99.41±0.02 | 99.14±0.01 | 99.08±0.06 | 99.04±0.09 | 99.11±0.04 | 95.33±0.63 | 93.43±0.09 | 84.79±0.38 |
| ANL-FL | 99.39±0.04 | **99.25±0.04** | 99.21±0.04 | **99.19±0.03** | **99.24±0.03** | 98.23±0.24 | 90.77±0.20 | 84.19±0.44 |
| ANL-CE-ER | 99.39±0.04 | 99.24±0.05 | 99.19±0.10 | **99.20±0.03** | **99.25±0.03** | 98.28±0.14 | 90.76±0.24 | 84.98±0.49 |
| ANN | 99.37 | 99.19 | 99.10 | 99.06 | 98.97 | 95.31 | 91.99 | 84.91 |
| WANN | 99.36 | **99.32** | **99.27** | **99.19** | **99.24** | 97.21 | 91.68 | 85.44 |

Table 2: Accuracy ($\uparrow$) on two *clean* and *noisy* datasets (Wei et al., 2022). The two best results are highlighted in **bold**. Note that NR stands for Noise Rate. CIFAR-10N includes three annotations per image, and *Noisy split* denotes various aggregation strategies. Following the proposed naming convention, *Aggr* refers to a majority voting, *R-i* ($i \in \{1, 2, 3\}$) denotes the $i$-th submitted label for each image, and *Worst* denotes the selection of only wrong labels, if applicable. Conversely, CIFAR-100N presents just one annotation per image, thereby offering one single noisy split.

Due to the limited dataset size, we further extract an *additional* noisy validation set of 250 samples (*i.e.*, 25 samples per class). However, WANN does not require this additional data, which is often difficult to obtain in limited-data settings.

Artificial noise is introduced in CIFAR-10 using traditional methods: symmetric (Patrini et al., 2017), asymmetric (Scott et al., 2013), and instance-dependent (Xia et al., 2020) noise. Symmetric noise involves the random switching of one label with any other label in the dataset. Asymmetric noise flips a specific label to another fixed, incorrect label (e.g., "deer" mislabeled as "horse"), simulating systematic class confusion. Finally, instance-dependent noise takes into account the relationships between the features within the dataset. Following Zhu et al. (2022), we report the accuracy on 60% symmetric noise, 30% asymmetric noise, and 40% instance-dependent noise. Following Ye et al. (2023), our asymmetric flips are defined as: TRUCK → AUTOMOBILE, BIRD → AIRPLANE, DEER → HORSE, CAT ↔ DOG. Due to the stochasticity in subsampling and noise generation, we average the results across five seed runs.

**Results** As shown in Table 3, across all experiments (*i.e.*, two sample sizes and three noise injections each), WANN is clearly superior to both ANN and robust loss functions. The advantage of adaptive methods over robust loss functions is particularly evident under heavy noise conditions (*e.g.*, 60% *sym.*). Although this performance gap narrows slightly as data size increases (*i.e.*, under less challenging conditions), the weighting scheme introduced in WANN continues to demonstrate clear benefits over ANN with minimal overhead. Moreover, a paired $t$-test ($p < 0.05$) conducted across five seed runs confirms that WANN is the *significantly* best method in all but two cases (*cf.* Table 3). Furthermore, with 100 samples per class, WANN consistently exhibits lower standard deviation across seed runs, thereby confirming its higher robustness against competing methods. Figure 3 illustrates that under asymmetric noise, incorrect labels may cluster closely in the embedding space. This in fact poses challenges for the traditional $k$-NN but is mitigated by WANN. Notably, while linear methods approach and surpass the performance of ANN with 100 samples per class, they do not outperform WANN.

| Pattern NR | CIFAR-10 (#50) | | | | CIFAR-10 (#100) | | | |
|---|---|---|---|---|---|---|---|---|
| | Clean - | Symmetric 60% | Asymmetric 30% | Instance 40% | Clean - | Symmetric 60% | Asymmetric 30% | Instance 40% |
| CE | $95.06_{\pm1.71}$ | $68.41_{\pm4.13}$ | $88.77_{\pm0.65}$ | $84.58_{\pm1.43}$ | $96.98_{\pm0.95}$ | $81.17_{\pm2.52}$ | $90.97_{\pm1.48}$ | $88.14_{\pm1.32}$ |
| FL | $94.66_{\pm1.55}$ | $68.19_{\pm3.24}$ | $88.30_{\pm1.09}$ | $81.41_{\pm1.85}$ | $96.85_{\pm0.81}$ | $80.33_{\pm0.68}$ | $89.89_{\pm1.00}$ | $85.88_{\pm1.74}$ |
| MAE | $95.68_{\pm0.84}$ | $83.98_{\pm4.41}$ | $93.31_{\pm1.11}$ | $92.41_{\pm2.23}$ | $97.64_{\pm0.22}$ | $93.33_{\pm1.96}$ | $95.55_{\pm0.74}$ | $94.96_{\pm1.00}$ |
| GCE | $96.25_{\pm0.65}$ | $82.35_{\pm3.15}$ | $92.22_{\pm1.08}$ | $\mathbf{93.20_{\pm0.86}}$ | $97.48_{\pm0.41}$ | $91.81_{\pm2.05}$ | $94.85_{\pm1.32}$ | $94.80_{\pm0.92}$ |
| SCE | $96.05_{\pm0.60}$ | $84.52_{\pm4.01}$ | $92.73_{\pm0.55}$ | $90.91_{\pm3.76}$ | $97.78_{\pm0.28}$ | $93.01_{\pm2.12}$ | $95.06_{\pm1.19}$ | $\mathbf{95.63_{\pm1.05}}$ |
| ELR | $95.69_{\pm0.73}$ | $85.26_{\pm3.99}$ | $93.02_{\pm2.24}$ | $90.38_{\pm3.97}$ | $96.93_{\pm0.53}$ | $91.18_{\pm4.06}$ | $95.32_{\pm0.85}$ | $94.75_{\pm1.25}$ |
| NCE-MAE | $96.19_{\pm0.76}$ | $83.54_{\pm3.54}$ | $92.84_{\pm0.70}$ | $92.45_{\pm2.23}$ | $97.75_{\pm0.32}$ | $93.02_{\pm1.32}$ | $94.96_{\pm1.32}$ | $95.18_{\pm1.07}$ |
| NCE-RCE | $96.00_{\pm0.58}$ | $83.92_{\pm4.24}$ | $93.28_{\pm1.08}$ | $91.18_{\pm4.13}$ | $97.65_{\pm0.21}$ | $93.32_{\pm1.94}$ | $95.57_{\pm0.70}$ | $95.03_{\pm1.10}$ |
| NFL-RCE | $96.00_{\pm0.58}$ | $83.92_{\pm4.24}$ | $93.28_{\pm1.08}$ | $91.18_{\pm4.13}$ | $97.65_{\pm0.21}$ | $93.31_{\pm1.95}$ | $95.57_{\pm0.70}$ | $95.03_{\pm1.10}$ |
| NCE-AGCE | $96.05_{\pm0.47}$ | $83.98_{\pm4.26}$ | $93.27_{\pm1.10}$ | $91.01_{\pm3.96}$ | $97.80_{\pm0.28}$ | $93.06_{\pm2.18}$ | $95.52_{\pm0.68}$ | $95.06_{\pm1.11}$ |
| NCE-AUL | $96.04_{\pm0.60}$ | $83.84_{\pm4.26}$ | $92.75_{\pm0.58}$ | $90.66_{\pm3.73}$ | $97.66_{\pm0.21}$ | $91.04_{\pm4.38}$ | $94.72_{\pm0.93}$ | $95.61_{\pm1.23}$ |
| NCE-AEL | $96.26_{\pm0.65}$ | $82.85_{\pm3.18}$ | $93.42_{\pm1.16}$ | $92.43_{\pm1.90}$ | $97.36_{\pm0.31}$ | $92.30_{\pm1.81}$ | $94.93_{\pm1.11}$ | $95.17_{\pm0.94}$ |
| ANL-FL | $93.32_{\pm2.02}$ | $78.54_{\pm2.80}$ | $91.15_{\pm1.78}$ | $87.84_{\pm1.93}$ | $96.90_{\pm0.56}$ | $87.66_{\pm3.55}$ | $92.79_{\pm1.30}$ | $91.30_{\pm2.66}$ |
| ANL-CE-ER | $95.16_{\pm0.99}$ | $76.20_{\pm6.63}$ | $91.16_{\pm1.50}$ | $86.79_{\pm2.09}$ | $96.59_{\pm1.00}$ | $90.36_{\pm1.05}$ | $93.55_{\pm1.63}$ | $92.44_{\pm0.70}$ |
| ANN | $97.42_{\pm0.33}$ | $\mathbf{92.26_{\pm1.69}}$ | $\mathbf{93.96_{\pm0.70}}$ | $92.63_{\pm2.37}$ | $98.24_{\pm0.23}$ | $\mathbf{95.83_{\pm0.49}}$ | $\mathbf{95.64_{\pm0.34}}$ | $95.60_{\pm0.85}$ |
| WANN | $97.29_{\pm0.38}$ | $\mathbf{93.08_{\pm2.28}}$ | $\underline{\mathbf{95.00_{\pm0.80}}}$ | $\mathbf{93.77_{\pm2.29}}$ | $98.18_{\pm0.18}$ | $\underline{\mathbf{96.79_{\pm0.42}}}$ | $\underline{\mathbf{96.92_{\pm0.25}}}$ | $\mathbf{96.84_{\pm0.40}}$ |

Table 3: Accuracy (↑) on limited noisy data settings. Both experiments are conducted on CIFAR-10, with 50 noisy samples per class (left) and 100 noisy samples per class (right). We bold the **top two methods** and underline the **significantly best** one if their difference is statistically significant (paired $t$-test, $p < 0.05$).

## 4.4 Limited real-world noisy data

We further assess classification performance on four stratified subsets of Animal-10N (Song et al., 2019) consisting of {500, 1000, 2500, 5000} total samples. It is a widely adopted dataset for label noise benchmarks (Ye et al., 2023) and includes 50,000 samples across 10 classes, with approximately $\approx 8\%$ noisy labels. We compare the performance of ANN and WANN against three representative loss functions (*cf.* Figure 4). As in Section 4.3, solely for linear probing methods, we extract an *additional* noisy validation set of 250 samples. Furthermore, we do not tune loss-specific hyperparameters for this dataset; instead, we used those found for CIFAR-10. We report the results obtained across five seed runs.

**Results** As shown in Figure 4, while linear probing approaches $k$-NN methods (*i.e.*, ANN and WANN) with larger subsets, its effectiveness diminishes with limited data. Specifically, when reduced to about 50 (noisy) samples per class, linear methods substantially underperform against ANN and WANN, exhibiting a larger 95% confidence interval. This performance gap narrows only when the sample size is increased to about 2,500 and 5,000 samples.

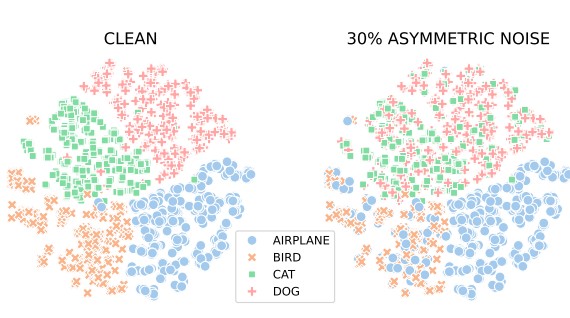

Figure 3: t-SNE projection of a CIFAR-10 subset and its noisy (30% asymmetric) counterpart.

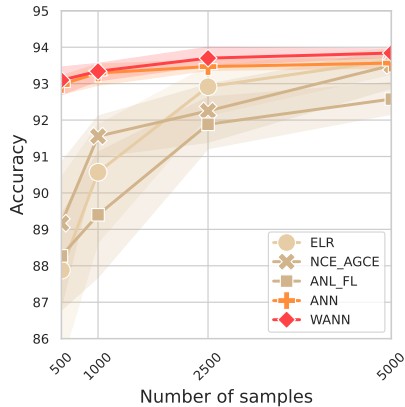

Figure 4: Average accuracy (↑) and 95% confidence interval on stratified subsets of Animal-10N.

### 4.5 Generalization to medical data

To demonstrate the generalization capabilities and effectiveness of `WANN`, we provide another pool of experiments on two challenging medical datasets, which are semantically far from the pre-training data of the employed foundation model. Specifically, we used two datasets from the MedMNIST+ dataset collection (Yang et al., 2023), namely BreastMNIST and DermaMNIST, each with resolution 224×224. BreastMNIST is a *binary* dataset with *only* 546 *training samples*. Given its binary nature, we limit our noise injection to symmetric noise at 20%, 30%, and 40% noise ratios. DermaMNIST, on the other hand, is a *multi-class* dataset with 7 classes and 7,007 training samples. Thus, we apply the same noise settings as previously (*cf.* Table 4). For asymmetric noise, we shift each class by one in a circular manner. We use the loss-hyperparameters defined for CIFAR-10 and report average accuracy over five runs.

**Results** Table 4 presents the results for the medical datasets. Notably, on BreastMNIST, `WANN` and `ANN` exhibit the highest accuracy with both clean and noisy training sets. Furthermore, `WANN` outperforms `ANN` with increased symmetric noise. While it might be expected that $k$-NN-based approaches would outperform linear methods on a binary problem with a small dataset, the weighting scheme in `WANN` demonstrates clear benefits on DermaMNIST as well, with a *significant* improvement on *instance dependent* label noise. In fact, although GCE outperforms `ANN` with symmetric noise, it does not surpass `WANN`, which consistently achieves higher accuracy and lower standard deviation, demonstrating greater robustness.

| | BreastMNIST | | | | DermaMNIST | | | |
|---|---|---|---|---|---|---|---|---|
| **Pattern NR** | Clean - | Symmetric 20% | Symmetric 30% | Symmetric 40% | Clean - | Symmetric 60% | Asymmetric 30% | Instance 40% |
| CE | $68.72_{\pm2.73}$ | $56.67_{\pm3.86}$ | $55.38_{\pm5.42}$ | $52.18_{\pm4.01}$ | $78.82_{\pm0.32}$ | $60.17_{\pm2.02}$ | $69.09_{\pm1.53}$ | $54.57_{\pm4.89}$ |
| FL | $67.82_{\pm5.32}$ | $59.49_{\pm4.43}$ | $57.05_{\pm6.18}$ | $53.33_{\pm5.40}$ | $78.80_{\pm1.66}$ | $60.20_{\pm3.56}$ | $67.36_{\pm1.31}$ | $52.69_{\pm5.10}$ |
| MAE | $66.67_{\pm2.26}$ | $56.03_{\pm8.39}$ | $53.72_{\pm8.30}$ | $53.59_{\pm6.34}$ | $73.38_{\pm0.21}$ | $63.16_{\pm3.45}$ | $66.55_{\pm1.87}$ | $58.76_{\pm2.68}$ |
| GCE | $70.13_{\pm2.24}$ | $61.03_{\pm5.34}$ | $55.64_{\pm7.27}$ | $51.03_{\pm4.82}$ | $74.16_{\pm1.21}$ | $\mathbf{67.16_{\pm2.26}}$ | $68.52_{\pm1.15}$ | $59.53_{\pm3.19}$ |
| SCE | $66.54_{\pm2.48}$ | $55.90_{\pm8.32}$ | $54.10_{\pm8.48}$ | $52.95_{\pm6.18}$ | $73.29_{\pm1.57}$ | $62.34_{\pm3.19}$ | $70.07_{\pm1.54}$ | $60.55_{\pm2.67}$ |
| ELR | $60.00_{\pm5.43}$ | $57.44_{\pm6.46}$ | $55.64_{\pm6.17}$ | $52.18_{\pm4.38}$ | $70.58_{\pm1.41}$ | $63.87_{\pm2.69}$ | $67.67_{\pm0.55}$ | $59.57_{\pm10.19}$ |
| NCE-MAE | $68.72_{\pm2.96}$ | $57.56_{\pm8.13}$ | $53.59_{\pm8.05}$ | $53.21_{\pm5.97}$ | $73.10_{\pm0.96}$ | $61.89_{\pm3.23}$ | $70.00_{\pm1.44}$ | $60.61_{\pm4.09}$ |
| NCE-RCE | $67.18_{\pm2.34}$ | $57.82_{\pm8.78}$ | $53.72_{\pm8.28}$ | $53.21_{\pm6.03}$ | $73.11_{\pm0.54}$ | $61.67_{\pm2.96}$ | $67.04_{\pm1.59}$ | $59.48_{\pm3.24}$ |
| NFL-RCE | $66.28_{\pm3.00}$ | $57.56_{\pm8.63}$ | $53.72_{\pm8.28}$ | $53.08_{\pm5.84}$ | $73.11_{\pm0.54}$ | $61.67_{\pm2.96}$ | $67.04_{\pm1.59}$ | $59.48_{\pm3.24}$ |
| NCE-AGCE | $66.28_{\pm2.49}$ | $58.72_{\pm8.28}$ | $53.59_{\pm8.50}$ | $53.72_{\pm6.14}$ | $73.85_{\pm1.36}$ | $61.58_{\pm3.06}$ | $66.63_{\pm1.51}$ | $59.48_{\pm3.21}$ |
| NCE-AUL | $66.67_{\pm2.40}$ | $57.56_{\pm7.06}$ | $53.72_{\pm8.02}$ | $52.69_{\pm5.54}$ | $72.90_{\pm0.37}$ | $61.63_{\pm2.91}$ | $67.21_{\pm1.06}$ | $60.10_{\pm3.87}$ |
| NCE-AEL | $66.67_{\pm3.44}$ | $59.74_{\pm7.89}$ | $55.77_{\pm8.44}$ | $53.46_{\pm5.99}$ | $73.52_{\pm0.93}$ | $61.11_{\pm2.76}$ | $70.00_{\pm1.78}$ | $59.36_{\pm2.75}$ |
| ANL-FL | $69.10_{\pm2.58}$ | $60.26_{\pm5.57}$ | $55.13_{\pm7.95}$ | $54.10_{\pm6.13}$ | $61.82_{\pm0.22}$ | $57.46_{\pm2.22}$ | $58.36_{\pm1.21}$ | $36.81_{\pm5.83}$ |
| ANL-CE-ER | $67.18_{\pm2.51}$ | $60.00_{\pm7.85}$ | $55.64_{\pm8.10}$ | $53.97_{\pm6.80}$ | $42.51_{\pm1.83}$ | $36.20_{\pm0.73}$ | $36.10_{\pm1.83}$ | $30.63_{\pm2.41}$ |
| ANN | 76.92 | $\mathbf{74.62_{\pm1.12}}$ | $\mathbf{69.62_{\pm1.79}}$ | $\mathbf{63.08_{\pm2.28}}$ | 73.77 | $65.87_{\pm0.49}$ | $\mathbf{\underline{70.12_{\pm0.78}}}$ | $\mathbf{62.81_{\pm1.73}}$ |
| WANN | 75.00 | $\mathbf{74.23_{\pm0.94}}$ | $\mathbf{72.69_{\pm1.38}}$ | $\mathbf{65.64_{\pm1.50}}$ | 73.07 | $\mathbf{69.96_{\pm0.23}}$ | $\mathbf{\underline{71.42_{\pm0.14}}}$ | $\mathbf{\underline{67.49_{\pm1.20}}}$ |

Table 4: Accuracy (↑) on BreastMNIST and DermaMNIST. We bold the **top two methods** and underline the **significantly best** one if their difference is statistically significant (paired $t$-test, $p < 0.05$).

### 4.6 Long-tailed noisy data

While $k$-NN is inherently robust, the selection of an appropriate value of $k$ is critical and not trivial, especially considering the unknown noise rate and potential class imbalances. In this experiment, we assess the performance of `ANN` and `WANN`, as compared to two fixed $k$-NN approaches. As discussed in the method section, both adaptive methods used $(k_{\min}, k_{\max}) = (11, 51)$. They are compared with their respective fixed $k$-NN counterparts, namely 11-NN and 51-NN. For benchmarking on long-tailed problems, we use CIFAR-10LT and CIFAR-100LT, following prior studies (Cao et al., 2019; Du et al., 2023), with imbalance ratios of 1% and 10%, respectively. It denotes the ratio between the least and most frequent class, with an exponentially decaying imbalance across all other classes. In CIFAR-10LT, the majority class has 5000 samples, while the minority class consists of 50 samples. Consequently, in CIFAR-100LT, we have 500 and 50 samples, respectively.

As in previous experiments, noise is artificially injected. To emphasize the generalizability across different noise patterns and severities, we inject symmetric noise ranging from 20% to 60%, asymmetric noise from 20% to 40%, and instance-dependent noise from 20% to 40%, reporting the accuracy over five runs.

**Results** The results presented in Figure 5 align with our expectations. Under limited noise conditions, a lower value of $k$ ($k = 11$) generally achieves higher performance, while a higher value ($k = 51$) shows greater robustness against more severe noise levels. In contrast, adaptive methods consistently approach the best performances across various noise rates and patterns. `WANN`'s weighting scheme exhibits higher gains compared to `ANN`, especially under heavy asymmetric and instance-dependent noise. Additionally, considering all experiments and respective seed runs (due to the dataset variations from the stochastic noise injection), `WANN` emerges as the *significantly best* method (Wilcoxon signed-rank test, $p < 0.05$).

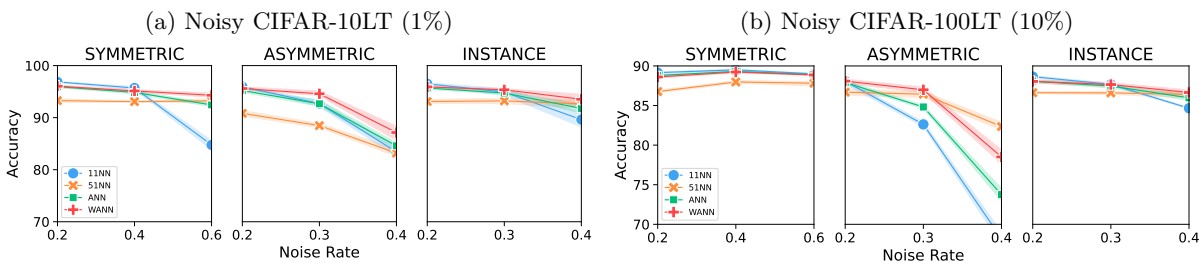

Figure 5: Accuracy of fixed and adaptive $k$-NN based approaches on CIFAR-10LT and CIFAR-100LT, with 1% and 10% imbalance ratios. The imbalance ratio denotes the ratio between the least and the most frequent class, with an exponentially decaying imbalance between all other classes. Notably, `WANN` is close to the best performance across any noise pattern and severity. Furthermore, `WANN` is the *significantly best* method across all experiments and datasets (Wilcoxon signed-rank test, $p < 0.05$).

### 4.7 Dimensionality reduction

To showcase the effectiveness of the proposed Filtered LDA (`FLDA`) approach, we evaluate `WANN`'s classification accuracy on the linearly projected test set, utilizing the complete projected training set as `WANN`'s support set. Four different projections are employed for this analysis. First, as a baseline, we report classification performances without dimensionality reduction, labeled as "None". Subsequently, we utilize Principal Component Analysis (PCA) with a fixed number of components, specifically 200, explaining an average of 77.19% of the total variance across each dataset. Additionally, we compare this with plain Linear Discriminant Analysis (LDA) fed with the entire noisy dataset. Finally, we present the effectiveness of our filtering approach before applying LDA, named as `FLDA`. To this extent, we recall that LDA projects the dataset along $C - 1$ axes, where $C$ is the number of classes. Consistently with our previous experimental settings, the accuracy is reported on datasets with 60% symmetric noise, 30% asymmetric noise, and, 40% instance-dependent label noise. The experiments cover three different datasets: CIFAR-10, CIFAR-100, and MNIST (Lecun et al., 1998). Following Ye et al. (2023), for asymmetric noise on CIFAR-100 we group the 100 classes into 20 super-classes and flip each class within its super-class in a circular fashion, while on MNIST we use the mapping $7 \to 1$, $2 \to 7$, $5 \leftrightarrow 6$, $3 \to 8$.

**Results** The results are visually reported in Figure 6. On the clean dataset, LDA and `FLDA` exhibit among the highest gains, substantially reducing the dimensionality and thereby reducing the overall sparsity. Here, LDA outperforms its *filtered* counterpart because of the absence of noisy labels. In fact, these performance gains vanish in the presence of label noise, confirming that `FLDA` is *significantly* better than all other dimensionality-reduction methods (Wilcoxon signed-rank test, $p<0.05$) under varying noise conditions. The reduction in dimensionality not only enhances classification performance but also contributes to faster computational speed and potential storage efficiency.

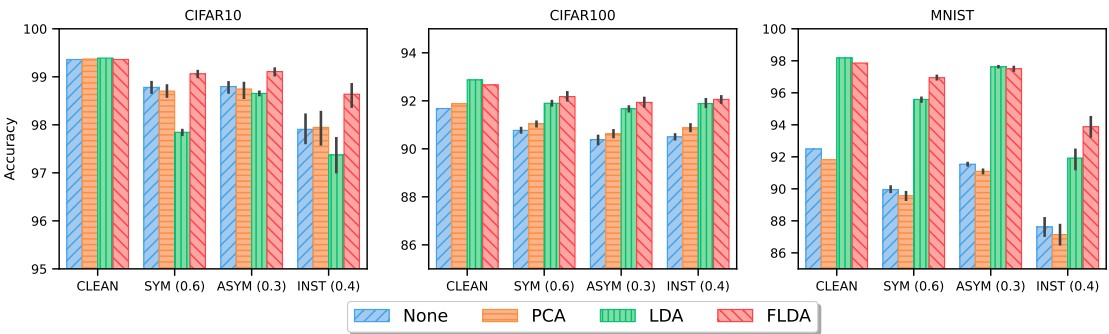

Figure 6: Classification accuracy of `WANN` under three dimensionality reduction strategies, evaluated across three datasets and three different noise settings each. *None* reflects the performance using the original image embeddings without dimensionality reduction. *PCA* and *LDA* show the classification performances on the linearly projected feature spaces, utilizing the whole noisy training set. Finally, `FLDA` involves the proposed filtering approach. `FLDA` is the *significantly best* dimensionality reduction method across all experiments and datasets (Wilcoxon signed-rank test, $p < 0.05$).

Notably, using DINOv2 ViT-L as feature extractor, we have an initial embedding size of 1024. Thus, we significantly improved the classification performance with 100× (CIFAR-10, MNIST) and 10× (CIFAR-100) smaller embeddings.

## 4.8 Explainability benefits

While deep learning methods are often viewed as black boxes with limited explainability, some traditional machine learning approaches inherently allow for a certain level of explainability. Notably, $k$-NN-based methods provide explainability through the representative nature of the neighborhood. Exploring the neighborhood for each prediction provides a deeper understanding of the factors contributing to correct or erroneous classifications. As illustrated in Figure 2, we present four test samples along with their top 3 closest training examples from STL-10 (Coates et al., 2011), a subset of ImageNet (Deng et al., 2009). In Figure 2a, we highlight two test images with clearly incorrect reference labels, supported by clean training samples. In contrast, Figure 2b shows two ambiguous test labels, including multiple known objects within a single image, such as a `bird` with a `ship` in the background or a `dog` playing with a `cat`. The pitfalls of ImageNet labels were already illustrated by Beyer et al. (2020). However, this provides a simple yet effective way to manually inspect potentially wrong labels within our datasets. Indeed, the explainability of $k$-NN-based approaches becomes particularly valuable in the context of active label correction (Rebbapragada et al., 2012; Bernhardt et al., 2022), where human intervention is employed to re-annotate noisy samples.

## 5 Discussion and conclusions

**Limitations** The effectiveness of $k$-NN relies on a representative feature representation, posing challenges for domains not represented during pre-training. However, this limitation can be mitigated by using alternative feature extractors. This is now possible thanks to the growing availability of such large open-source models. Moreover, while the proposed `FLDA` significantly enhances classification performance, it requires a sufficient sample size to generate reliable projection vectors. Finally, to mitigate the computational demands of $k$-NN's exhaustive search, established *Approximate*-NN methods (Malkov & Yashunin, 2020; Guo et al., 2020; Johnson et al., 2021) provide an effective solution, achieving comparable classification performance with substantially lower computational complexity. Indeed, we could expect a negligible influence on classification performance, as distance metrics inherently only approximate the nearest sample.

**Side benefits and generalization**   The efficiency of our classification method, further optimized with dimensionality reduction, enables its deployment with limited resources. Moreover, the reliable feature space of current feature extractors becomes particularly suitable for $k$-NN-based approaches, possibly generalizing to different domains. For instance, we observed outstanding classification performance and robustness in the medical domain, where interpretability, data scarcity, and data imbalance are still open challenges. Furthermore, the compliance with data regulations, such as the *right to be forgotten* (Mantelero, 2013), is facilitated by the proposed database approach. The flexibility to remove or add data points without extensive re-training or fine-tuning, aligns with privacy guidelines, providing a cost-effective solution. Lastly, our method can be readily applied to any domain, tabular or otherwise, as long as we can obtain a meaningful semantic space. While our current work focuses on image embeddings derived from large vision foundation models, the same concept extends to language, audio, or multimodal embeddings, assuming they capture the semantic relationships necessary for defining reliable neighborhoods.

**Conclusion**   We introduce `WANN`, a Weighted Adaptive Nearest Neighbor approach, based on a *reliability score* quantifying the correctness of the training labels. Our experiments demonstrate its higher robustness with respect to label noise, particularly in scenarios with limited labeled data, surpassing traditional models relying on robust loss functions. Additionally, the incorporation of dimensionality reduction techniques enhances its performance under noisy labels, facilitated by the proposed weighted scoring.

### Acknowledgments

This study was funded through the Hightech Agenda Bayern (HTA) of the Free State of Bavaria, Germany.

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

# A  Generalizability across backbones

This section illustrates two additional experiments using a ViT-L backbone pretrained on ImageNet in a supervised fashion. We compared our method with the same subset of loss functions utilized in Section 4.4 (CE, ELR, NCE-AGCE, ANL-FL, ANL-CE-ER). On CIFAR-N (*c.f.* Table A.1), while `WANN` shows slightly lower performance than the best-performing robust loss function on this particular backbone, our `FLDA` approach offers a substantial performance boost, achieving results comparable to or even surpassing those of the robust loss functions.

Additionally, we conducted an experiment analogous to Section 4.3 using CIFAR-10 with subsets of 50 and 100 samples per class (*c.f.* Table A.2). Notably, when using only 50 samples per class, the adaptive methods substantially outperform the robust loss functions. These findings demonstrate that the advantages of our approach are not merely due to the DINOv2 backbone's inherent strengths but rather underscore the general applicability and robustness of our method.

| Noisy split | CIFAR-10N | | | | | | CIFAR-100N | |
|---|---|---|---|---|---|---|---|---|
| | Clean | Aggr. | R1 | R2 | R3 | Worst | Clean | Noisy |
| NR ($\approx$) | - | 9.01% | 17.23% | 18.12% | 17.64% | 40.21% | - | 40.20% |
| CE | 96.56±0.06 | 95.19±0.09 | 94.96±0.27 | 95.01±0.16 | 95.03±0.15 | 91.35±0.16 | 84.66±0.30 | 74.48±0.10 |
| ELR | 96.37±0.09 | **95.94±0.07** | 95.74±0.14 | 95.85±0.07 | 95.71±0.38 | 94.25±0.35 | 84.03±0.06 | **76.50±0.28** |
| NCE-AGCE | 96.50±0.06 | **96.10±0.07** | **95.96±0.06** | **95.98±0.10** | **95.93±0.09** | 93.79±0.25 | 84.62±0.17 | **77.92±0.09** |
| ANL-FL | 96.09±0.19 | 95.90±0.10 | 95.60±0.23 | 95.78±0.10 | 95.45±0.27 | **94.41±0.12** | 79.41±0.33 | 71.24±0.32 |
| ANL-CE-ER | 96.18±0.14 | 95.78±0.26 | 95.52±0.35 | 95.63±0.32 | 95.72±0.16 | **94.79±0.15** | 79.57±0.30 | 73.07±0.26 |
| ANN | 94.97 | 94.50 | 94.25 | 94.05 | 94.19 | 90.53 | 78.31 | 70.29 |
| WANN | 94.64 | 94.36 | 94.15 | 94.17 | 94.25 | 92.22 | 77.61 | 70.95 |
| WANN+FLDA | 95.99 | 95.81 | **95.77** | **95.86** | **95.90** | 92.94 | 81.93 | 75.15 |

Table A.1:  Accuracy on CIFAR-N datasets utilizing a ViT Large pre-trained on ImageNet. The results are averaged over five seed runs.

| Pattern | CIFAR-10 (#50) | | | | CIFAR-10 (#100) | | | |
|---|---|---|---|---|---|---|---|---|
| | Clean | Symmetric | Asymmetric | Instance | Clean | Symmetric | Asymmetric | Instance |
| NR | - | 60% | 30% | 40% | - | 60% | 30% | 40% |
| CE | 95.23±0.67 | 74.46±2.13 | 88.07±1.45 | 81.76±1.96 | 95.95±0.46 | 83.87±1.00 | 90.92±0.77 | 86.89±2.02 |
| ELR | 95.00±0.34 | 82.40±4.65 | 93.63±0.63 | 89.98±4.15 | 96.45±0.49 | 94.02±1.90 | **94.14±2.14** | 94.32±1.81 |
| NCE-AGCE | 95.49±0.97 | 85.94±2.62 | 93.22±1.13 | 90.82±3.19 | 96.93±0.35 | 94.71±0.96 | **95.66±0.57** | **95.40±0.84** |
| ANL-FL | 94.39±1.08 | 78.00±2.31 | 88.02±2.77 | 82.80±2.15 | 96.26±0.70 | 90.54±1.40 | 92.10±0.65 | 88.79±2.24 |
| ANL-CE-ER | 94.82±0.89 | 78.18±3.90 | 90.05±0.71 | 83.13±4.20 | 95.99±0.76 | 90.54±0.97 | 92.97±1.06 | 89.20±1.89 |
| ANN | 97.57±0.14 | **94.00±0.62** | **94.48±0.82** | 93.75±1.79 | 88.70±1.32 | **96.74±0.37** | 92.37±1.19 | 94.47±1.66 |
| WANN | 97.58±0.15 | **95.47±0.68** | **95.32±0.74** | **95.22±1.19** | 88.69±1.31 | **97.10±0.36** | 92.67±1.26 | **94.85±1.55** |

Table A.2: Accuracy on limited noisy data (50 and 100 samples per class) utilizing a ViT Large pre-trained on ImageNet. The results are averaged over five seed runs.

# B  Robustness of reliability score

## B.1  Signal for linear classifiers

This experiment aims to evaluate the utility and robustness of our reliability score. The hypothesis is the following: *if our reliability score is a robust indicator of label noise, a linear layer should be able to utilize this information just as effectively as k-NN.* To test this hypothesis, we precomputed the reliability score for each training sample and concatenated it with the respective feature embedding. For test samples, we utilized an average reliability score from their training neighborhood, based on the intuition that the spatial distribution of these scores is both informative and reliable.

As illustrated in Table B.1, for CIFAR-10 and CIFAR-100, this straightforward approach of augmenting the feature embeddings with the reliability score slightly improves performance while further reducing the variability (lower standard deviation across five seed runs). These results hold promise that the reliability score provides a valuable signal that a linear classifier can utilize as effectively as the *k*-NN approach.

| Pattern NR | CIFAR-10 | | | | CIFAR-100 | | | |
|---|---|---|---|---|---|---|---|---|
| | Clean
- | Symmetric
60% | Asymmetric
30% | Instance
40% | Clean
- | Symmetric
60% | Asymmetric
30% | Instance
40% |
| CE | 99.40±0.03 | 98.00±0.13 | 96.47±0.19 | 95.66±0.72 | 78.82±0.32 | 60.17±2.02 | 69.09±1.53 | 54.57±4.89 |
| $\eta$-CE | 99.36±0.02 | 98.10±0.13 | 96.55±0.31 | 95.94±0.51 | 80.22±1.71 | 63.48±2.01 | 70.11±1.18 | 60.04±3.29 |

Table B.1: Accuracy on CIFAR-10 and CIFAR-100 under three label noise settings. We compare the standard performance using the raw embeddings (CE) and the ones concatenated with our reliability score ($\eta$-CE). The results are averaged over five seed runs.

## B.2 Distribution of the reliability scores

Utilizing a toy dataset consisting of $1,000$ samples and $40\%$ symmetric noise, we illustrate in Figure B.1: (top-left) the real class distribution, (top-right) the noisy class distribution, (bottom-left) the reliability scores assigned to each point, and (bottom-right) the distribution of reliability scores for clean vs. noisy samples. From the bottom-right plot, it is possible to observe that clean samples (blue) tend to have higher reliability scores, whereas noisy samples (red) cluster at lower or intermediate values. Points near decision boundaries or mislabeled neighbors also receive moderate or low reliability scores. This observation supports our intuition that WANN utilizes local consistency in the embedding space to effectively distinguish clean from noisy samples.

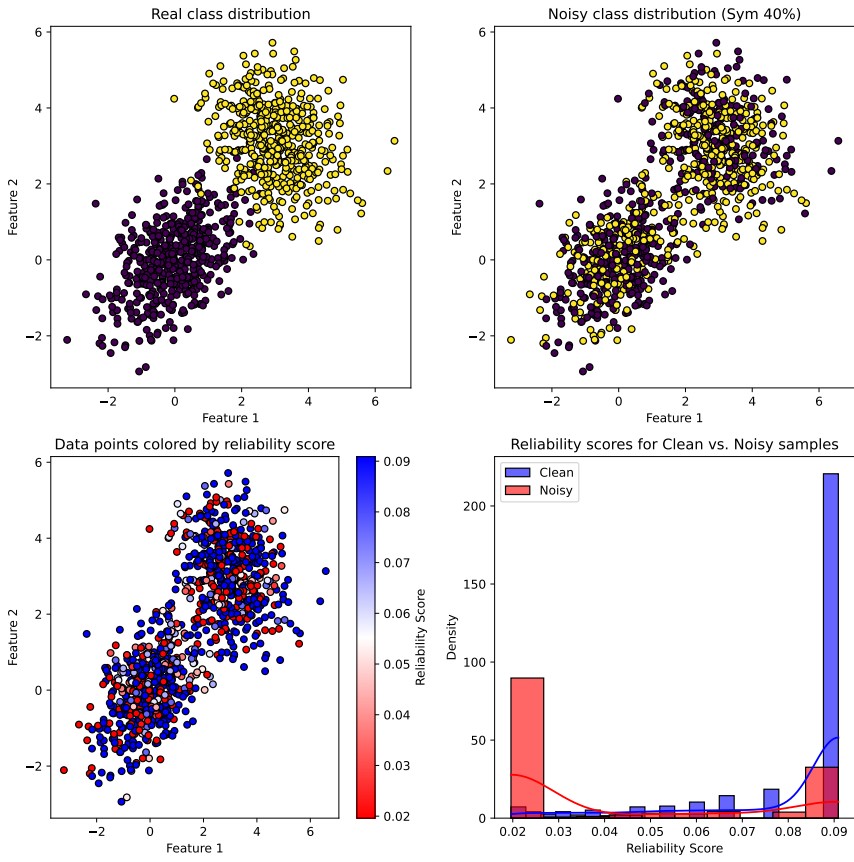

Figure B.1: Reliability score distribution on a toy dataset with $40\%$ symmetric noise.

