# OpenReview forum: "An Embedding is Worth a Thousand Noisy Labels"
_TMLR — Accepted by TMLR_

### Review · Reviewer_XUqU · 2025-02-12

**Summary Of Contributions:**

This paper investigates the noisy label problem from a novel perspective. Unlike existing works that primarily focus on training robust models, this study explores robust inference using self-supervised learning models. It introduces a method called Weighted Adaptive Nearest Neighbor (WANN), which assigns a reliability score to each training sample, where potentially noisy labels receive lower scores. During inference, predictions are derived from multiple local neighbourhoods, weighted by these reliability scores. Experimental results demonstrate that the proposed method is highly effective, particularly when using DINOv2 as the feature extractor.

**Audience:**

Yes

**Broader Impact Concerns:**

There are no ethical implications.

**Claims And Evidence:**

Yes

**Requested Changes:**

Please refer to the Strengths And Weaknesses section.

**Strengths And Weaknesses:**

Strengths:
- The paper explores a novel and interesting direction by addressing the label noise problem at the inference stage rather than during training.
- The proposed method, WANN, is technically sound and well-motivated.
- The evaluation is comprehensive, including existing baseline methods, which clearly demonstrate the improvement and validate the effectiveness of the proposed approach.

Weakness:
- The proposed method lacks novelty, as it can be seen as a weighted KNN approach for self-supervised learning (SSL) models. Existing SSL methods already utilize KNN for inference in classification tasks, and WANN is essentially a weighted extension, which feels somewhat intuitive rather than a fundamentally new contribution. The paper’s presentation does not provide much insight beyond the method itself, making it less engaging for readers.
- The proposed Filtered LDA also lacks significant novelty, as it is essentially a straightforward combination of filtering with LDA. The contribution feels incremental rather than introducing a new approach.
- The effectiveness of WANN is heavily dependent on representation quality, particularly when using DINOv2. The empirical improvements rely on this specific feature extractor, raising concerns about the method’s generalizability. The reviewer suggests a deeper analysis of why DINOv2 is necessary and why other methods listed in Table 1 are insufficient. This analysis would strengthen the contribution and provide deeper insights.
- It would be good to include SimCLR, BYOL, and VICReg in the analysis.
- All experiments are conducted on small-scale datasets such as CIFAR and MNIST, which limits the ability to assess the generalization of the proposed method. The reviewer suggests including evaluations on larger datasets, such as ImageNet.
- For real-world analysis, it would be more beneficial to explore some real-world medical datasets, such as the Cholec80 Tool classification [1]. SSL models are also trained for such datasets [2].

[1] Twinanda, Andru P., et al. "Endonet: a deep architecture for recognition tasks on laparoscopic videos." IEEE transactions on medical imaging 36.1 (2016): 86-97.\
[2] Zhou, Yuning, et al. "DDA: Dimensionality Driven Augmentation Search for Contrastive Learning in Laparoscopic Surgery." Medical Imaging with Deep Learning 2024.

---

> ### Author Response · Authors · 2025-03-07
> **Response to reviewer XUqU (1/2)**
>
> > The proposed method lacks novelty, as it can be seen as a weighted KNN approach for self-supervised learning (SSL) models.
> > The proposed Filtered LDA also lacks significant novelty.
>
> We appreciate and agree with the reviewer's observation that our method is grounded in well-established principles. Nevertheless, our work represents a broader paradigm shift: rather than training large networks from scratch, we utilize high-quality feature representations from foundation models and employ a simple, interpretable, and training-free classification scheme in the embedding space. This design choice directly addresses practical challenges such as mitigating overfitting to noisy labels, enabling transparent predictions, and simplifying dataset updates. The latter aspect is especially pertinent to the "right to be forgotten", which has motivated the current research into model unlearning. Indeed, under our offline strategy, unlearning (or learning new) samples is as straightforward as removing (or adding) entries in our vector database, eliminating the need for full model retraining.
>
>
> > The effectiveness of WANN is heavily dependent on representation quality, particularly when using DINOv2.
>
> We thank the reviewer for the opportunity to further emphasize this point. In fact, we argue that DINOv2 is not strictly necessary, but it does yield the best performance among the tested backbones. As shown in our  Table 1, and more extensively in the DINOv2 reference paper, it outperforms alternative architectures across various datasets and tasks. Furthermore, in our response to reviewer y5QB13 and in the now introduced Appendix A, we demonstrate that switching to an ImageNet-pretrained ViT-L still gives comparable or superior results compared to robust loss functions. Therefore, we confirm that our superior performance is not motivated by the implicit choice of DINOv2.
>
>
> > It would be good to include SimCLR, BYOL, and VICReg in the analysis.
>
> Following the reviewer's suggestion to evaluate other SSL backbones, we attempted to obtain suitable off-the-shelf checkpoints. We conducted an additional experiment to confirm on two backbones whether supervised training yields superior feature embeddings as compared to the proposed SSL approaches. In particular, we conduct this comparison to build a WANN classifier for CIFAR10. As feature extractors, we utilized a ResNet18 pretrained on STL-10 with BYOL [1] and SimCLR [2], and we also tested ResNet50  pretrained with SimCLR [3] on ImageNet.
>
> As reported in the Table below, WANN's performance on these embeddings remains lower than the one on standard ImageNet supervised pretraining. The gap is notably smaller for ResNet18, likely because STL-10 (used in its pretraining) shares the same 10 classes as CIFAR-10. Lastly, it is worth noting that ResNet50 pre-trained with SimCLR substantially underperforms its supervised counterpart; this is likely due to the increased complexity introduced by the larger-scale pretraining, especially compared to smaller-scale pretraining on STL-10.
>
> | **Architecture** | **Pretraining**       | **Accuracy** |
> |------------------|-----------------------|--------------|
> | ResNet18         | BYOL (STL-10)         | 72.43        |
> | ResNet18         | SimCLR (STL-10)       | 71.87        |
> | ResNet18         | Supervised (ImageNet) | 80.56        |
> | ResNet50         | SimCLR (ImageNet)     | 48.99        |
> | ResNet50         | Supervised (ImageNet) | 84.09        |
>
> [1] https://github.com/sthalles/PyTorch-BYOL
>
> [2] https://github.com/sthalles/SimCLR
>
> [3] https://github.com/AndrewAtanov/simclr-pytorch

---

> ### Author Response · Authors · 2025-03-07
> **Response to reviewer XUqU (2/2)**
>
> > The reviewer suggests including evaluations on larger datasets, such as ImageNet, and more real-world medical datasets, such as the Cholec80 Tool classification.
>
> We thank the reviewer for the recommendation of these extra datasets. For time reasons, we focused on the Cholec80 Tool classification (subset from [4] ), including five surgical scenes.  We select three scenes for training (41, 42, 43) and two scenes for testing (44, 45). Notably, the training dataset comprises 7 classes and is heavily imbalanced, with the following distribution (class: samples): {$3: 6655, 4: 5085, 5: 924, 1: 760, 0: 544, 6: 218, 2: 174$}. To manage this imbalance and obtain more meaningful test results, we grouped classes with fewer than 1000 samples into a single "OOD" class.
>
> As shown in the additional Table below, using the suggested self-supervised ResNet50 [5], WANN achieves higher accuracy and comparable balanced accuracy. When using DINOv2-Large, WANN again exhibits superior performance in terms of accuracy but slightly lower performance in balanced accuracy. Although this dataset is not a dedicated label noise benchmark, our offline k-NN strategy performs on par with or better than the online strategy, confirming its effectiveness.
>
> | **Architecture**      | **Accuracy**     | **Balanced Accuracy** |
> |-----------------------|------------------|-----------------------|
> | **ResNet50 (SimCLR)** |                  |                       |
> | CE                    | 57.72 $\pm$ 0.96 | 55.97 $\pm$ 0.52      |
> | WANN                  | 60.92            | 54.72                 |
> | **DINOv2 Large**      |                  |                       |
> | CE                    | 64.95 $\pm$ 0.26 | 65.47 $\pm$ 0.16      |
> | WANN                  | 66.74            | 62.44                 |
>
> [4] Abdulbaki Alshirbaji, Tamer, et al. "Cholec80-Boxes: Bounding-Box Labels for Surgical Tools in Five Cholecystectomy Videos" Data. (2025).
>
> [5] Zhou, Yuning, et al. "DDA: Dimensionality Driven Augmentation Search for Contrastive Learning in Laparoscopy Surgery." MIDL (2024).

---

### Review · Reviewer_y5QB · 2025-02-13

**Summary Of Contributions:**

The authors introduce a classification algorithm and a dimensionality reduction method designed to mitigate the effects of noisy labels. Their classification approach leverages self-supervised features from the DINOv2 foundation model and employs an adaptive nearest neighbor strategy to compute a reliability score using neighborhood information. Each new instance is then classified based on the reliability score of its closest training neighbor. The dimensionality reduction method is essentially LDA, but they filter out instances with low reliability scores to mitigate the effects of noisy labels.

**Audience:**

Yes

**Claims And Evidence:**

No

**Requested Changes:**

See Weaknesses above.

**Strengths And Weaknesses:**

Strengths:
- The approach is straightforward and intuitive.
- Extensive experiments across diverse settings empirically demonstrate its effectiveness.
- The method is computationally efficient.
- The paper is well-written and easy to follow.

Weaknesses (sorted descending by importance):
- **Comparison to related work:** While the paper builds on well-established nearest-neighbor approaches for noisy label learning, its positioning within the broader literature remains unclear. The authors note that previous k-NN methods have struggled with issues like a lack of explainability, data efficiency, and generalizability. However, the paper does not explicitly detail how their approach overcomes these specific challenges. *Moreover, the work of Zhu et al. (2022), which applied a k-NN strategy on the CLIP model, is highlighted as closely related but not sufficiently differentiated from the current approach.* The paper should clearly outline what makes their method distinct—whether it is through a novel mechanism for computing reliability scores, or improvements in explainability and data efficiency. Without this clear differentiation, it is difficult to evaluate the novelty and significance of the contribution relative to existing methods.
- **The choice of the foundation model:** The authors employ the DINOv2 model as the backbone, justifying their choice by evaluating the performance of their methods on different backbones. However, since the backbone is a crucial element of the strategy, the choice of DINOv2 appears suboptimal, and additional backbones should be considered. This concern arises from the fact that the DINOv2 paper explicitly optimizes their SSL model for k-NN performance, which might boost the observed performance of the approach explicitly on this backbone. Therefore, to ensure that the results are not merely a byproduct of the backbone's inherent strengths, it is necessary to evaluate the method using at least one other backbone. Additionally, incorporating a backbone for other modalities, such as text, would enhance the study’s generality. Otherwise, the title should be adjusted to reflect the focus on image-based analysis.
- **End-to-end finetuning:** Since the approach centers on a k-NN method, the model cannot be fine-tuned end-to-end, limiting training to frozen embeddings. This is problematic because, in practical applications, a perfect foundation model is rarely available, and there is often a need to optimize earlier layers. Furthermore, it would be insightful to explore the effectiveness of incorporating the reliability scores with a linear layer. If the reliability score is truly a robust indicator of noisy labels, a linear layer should be able to leverage this information just as effectively as the k-NN approach. Therefore, an experiment combining the reliability scores with a linear classifier would be a valuable addition.
- **Normalized feature embeddings:** The paper states that embeddings are normalized. However, because the SSL backbone inherently scales features by their importance, normalization might strip away critical information. Did you try experiments using unnormalized embeddings? Preserving the original scaling could potentially lead to better performance.
- **Hyperparameter tuning:** What is the reason reason that no loss-specific hyperparameters where optimized in 4.4?
- **Oversized figure 1:** The size of Figure 1 (the GA) appears excessive. I recommend reducing its scale and relocating it to the end of the introduction, as this adjustment would enhance the overall presentation of the paper.

---

> ### Author Response · Authors · 2025-03-07
> **Response to reviewer y5QB**
>
> > Comparison to related work.
>
> We thank the reviewer for the comment on our contributions. Unlike previous k-NN-based methods that primarily enhance DNN performance, our approach focuses on achieving robust downstream classification using a simple, interpretable, and generalizable k-NN framework. In contrast, Zhu et al. present a training-free solution that detects corrupted labels via local voting and ranking-based filtering. This was evaluated mainly by the F1-score for noise detection and supported by one experiment on enhanced downstream (DNNs) performance after cleaning. Our work demonstrates that our straightforward strategy can deliver robust downstream results without the need for data cleaning and subsequent downstream training. We now revisit this subsection to make our contributions clearer.
>
> >  To ensure that the results are not merely a byproduct of the backbone's inherent strengths, it is necessary to evaluate the method using at least one other backbone.
>
> We thank the reviewer for suggesting an additional ablation study to further validate our claims. Accordingly, we conducted two additional experiments using an ImageNet-pretrained ViT-L in a supervised fashion. We compared our method with the same subset of loss functions utilized in Section 4.4 (CE, ELR, NCE-AGCE, ANL-FL, ANL-CE-ER).
>
> In our experiments on CIFAR-N (c.f. additional Table A.1, Appendix A), while WANN shows slightly lower performance than the best-performing robust loss function on this particular backbone, our fLDA approach offers a substantial performance boost, achieving results comparable to or even surpassing those of the robust loss functions.
>
> Additionally, we conducted an experiment analogous to Section 4.3 using CIFAR-10 with subsets of 50 and 100 samples per class (c.f. additional Table A.2, Appendix A). Notably, when using only 50 samples per class, the adaptive methods substantially outperform the robust loss functions. These findings demonstrate that the advantages of our approach are not merely due to the DINOv2 backbone's inherent strengths but rather underscore the general applicability and robustness of our method.
>
> > It would be insightful to explore the effectiveness of incorporating the reliability scores with a linear layer.
>
> We thank the reviewer for the insightful suggestion. To test the hypothesis that the reliability score is a robust indicator of noisy labels, we performed an additional experiment by incorporating the reliability scores into a linear classifier. In our initial prototype, we precomputed the reliability score for each training sample and concatenated it with the respective feature embedding. For test samples, we utilized an average reliability score from their training neighborhood, based on the intuition that the spatial distribution of these scores is both informative and reliable.
>
> As shown in the additional Table B.1 (Appendix B1), for CIFAR-10 and CIFAR-100, this straightforward approach of augmenting the feature embeddings with the reliability score slightly improves performance while further reducing the variability (lower standard deviation across five seed runs). These results hold promise that the reliability score provides a valuable signal that a linear classifier can utilize as effectively as the k-NN approach. Therefore, future work may explore the design of robust loss functions guided by this informative noisy signal, as also highlighted by reviewer JgxJ.
>
> > Additionally, incorporating a backbone for other modalities, such as text, would enhance the study’s generality.
>
> We very much appreciate the suggestion of evaluating other modalities and can see this as a promising future direction.
> We have added this reference to our conclusions.
>
> >  Normalization might strip away critical information. Did you try experiments using unnormalized embeddings? Preserving the original scaling could potentially lead to better performance.
>
> We appreciate the suggestion. However, in our experiments, we evaluated both normalized and un-normalized embeddings and did not observe noticeable gains, possibly due to the relatively large neighborhoods.
>
> > What is the reason that no loss-specific hyperparameters were optimized in 4.4?
>
> Since our method is training-free and requires no hyperparameter tuning, we chose to maintain consistency by using the default hyperparameters recommended in a reference paper [1] for the competitor methods. Specifically for Animal10N, since no hyperparameters were specified, we adopted the settings proven effective on CIFAR-10, given their similar number of classes and overall data scale. This ensures a fair comparison while avoiding extensive hyperparameter searches.
>
> > Oversized Figure 1
>
> We thank the reviewer for the styling suggestion. Following your feedback, we have reduced the figure's size and placed it at the end of the introduction.
>
> [1] Ye, Xichen, et al. "Active negative loss functions for learning with noisy labels." NIPS  (2023)

---

### Review · Reviewer_JgxJ · 2025-02-24

**Summary Of Contributions:**

The authors proposed a learning algorithm which is robust to noisy labels. The novel part is to propose
the reliability score based on the nearest neighbor method on the embedding space.

**Audience:**

Yes

**Broader Impact Concerns:**

No concern.

**Claims And Evidence:**

Yes

**Requested Changes:**

Please, add intuitive explanations why the proposed method is better than existing algorithms.

**Strengths And Weaknesses:**

Stregnths: The algorithm is easy to implement and shows suprior performance compared to existing algorithms.

Weakness: It is unclear why the proposed algorithm is superior even thouhg the idea is simple. Below, I list up my comments and questions.

1. What happens when there is no $k$ satisfying the Eq (1)? I think that this situation could occur frequently for
noisy-labeled data in the minor class. I

2. In the last paragraph of page 4, $k$ would be $\mathcal{H}(x,y,\mathcal{X})$ (or $eta$). $k$ is an argument used to define $\mathcal{H}(x,y,\mathcal{X})$.

3. I understand the idea of using the smallest number of nearest neighbors for correct classification (i.e. k) for measuring the reliability
but I am not sure that 1/k is an optimal choice. For me, 1/k seems to be too naive.

4. I understand the intuition of (4) , but it seems too naive. Are there any theoretical justifications for (4)?
I worry that (4) depends only on the nearest datum but not on multiple nearest neighbors.

5. As far as I understand, 'Filtered dimensionality reduction' is needed since nearest neighbor methods are susceptible to the curse of dimensionality. I wonder how sensitive the nearest neighbor method to the dimension of data when obtaining the reliability scores. I think
that nearest neighbor is still sensitive and thus the reliability scores may not be informative when the dimension of data is high.

6. In numerical studies, it seems that WANN outperforms other methods using specially designed loss functions for noisy-labels.
However, I could not understand intuitively why WANN is better? More detailed explanations would be necessary.

7. Is it possible to combine the reliability score and specially designed loss function? For example, we could learn a prediction model
by minimizing the weighted empirical average of the specially designed loss functions where the weights are proportional to the proposed reliability scores. This idea would be promising since the nearest neighbor method is difficult to be deployed.

8. To justify the proposed method (e.g. explain why WANN is better than existing loss-based methods), I recommend
the authors to investigate how WANN works for a simple toy model such as two Gaussian mixture model.

9. I think that WANN can be applied to tabular data without modification. Is there any reason to focus only on classification of embedding vectors?

---

> ### Author Response · Authors · 2025-03-07
> **Response to reviewer JgxJ (1/2)**
>
> > What happens when there is no $k$ satisfying the Eq (1)?
>
> We appreciate the comment and apologize for not making it clear enough. In cases where no $k$ within the range $[k_{\text{min}},k_{\text{max}}]$ results in a correct prediction, the reliability score is assigned as the lower bound: i.e., $\frac 1 {k_{\text{max}}}$, as mentioned in Algorithm (1). We now clarify this edge-case throughout the text.
>
> > In the last paragraph of page 4, $k$ would be $\mathcal{H}(x,y,\mathcal{X})$. $k$ is an argument used to define $\mathcal{H}(x,y,\mathcal{X})$.
>
> We appreciate the detailed feedback. There was indeed a minor inconsistency in the notation. In Equation (1), the variable $k \in [k_{\text{min}},k_{\text{max}}]$ is used to indicate the number of neighbors considered, and it is not itself the reliability score. The reliability score is defined as $\eta = \frac 1 k$, where $k$ is the smallest value for which the k-NN prediction is correct. To clarify this, we have revised the notation by replacing $k$ with $k'$ when referring to the variable within the range $ [k_{\text{min}},k_{\text{max}}]$.
>
> > I am not sure that 1/k is an optimal choice. For me, 1/k seems to be too naive. I worry that (4) depends only on the nearest datum but not on multiple nearest neighbors.
>
> We thank the reviewer for the thoughtful questions. While the use of $\frac 1 k$ may appear naive at first glance, it is grounded in the intuition that clean samples are typically well-clustered in the embedding space, requiring only a small number of neighbors to yield a correct classification. Regarding Equation (4), although it relies on the nearest datum (or the smallest neighborhood size), it serves as a first-order approximation of local reliability. This simplicity provides both interpretability and ease of computation, which are crucial advantages in practical applications. That said, we acknowledge that more complex formulations, such as incorporating local density or point-wise distances, could refine this measure. However, accurately modeling densities and distances in high-dimensional spaces can be challenging, potentially adding significant complexity without guaranteeing proportional improvements.
>
> > I wonder how sensitive the nearest neighbor method to the dimension of data when obtaining the reliability scores.
>
> We appreciate the question and understand your concern regarding the curse of dimensionality. Indeed, nearest neighbor methods can be sensitive to high-dimensional data, which might, in turn, affect the reliability scores. However, we argue that by considering a sufficiently large neighborhood, this sensitivity is mitigated to a certain extent. In our experiments, we ran a sensitivity test using noise settings of symmetric 0.6, asymmetric 0.3, and instance-dependent 0.4 on both CIFAR-10 and CIFAR-100. We compared the full embedding size (1024) with two reduced versions obtained via PCA (512 and 128 components). We observed that the standard deviations of the reliability scores ranged from 0.04 to 0.13 for CIFAR-10 and from 0.16 to 0.73 for CIFAR-100. The higher variability on CIFAR-100 can also be attributed to the information loss induced by PCA.
>
> > I could not understand intuitively why WANN is better?
>
> We thank the reviewer for the opportunity to further emphasize this point. Intuitively, WANN tends to outperform robust loss functions because neural networks often overfit to label noise during training, whereas our offline k-NN approach avoids this issue by operating on fixed, high-quality embeddings from foundation models. This offline strategy not only mitigates overfitting but also offers significant interpretability and efficiency benefits.
>
> > Is it possible to combine the reliability score and specially designed loss function?
>
> We thank the reviewer for the insightful comment. As mentioned in our response to reviewer y5QB13, we observed slight gains with a naive implementation (i.e., concatenating the embedding and reliability score and training with a CE loss). We included this experiment in Appendix B.1. The results suggest that integrating our reliability score into specially designed loss functions is a promising direction, and we agree that further exploration of such hybrid models could be valuable, especially given the challenges associated with deploying k-NN methods in practice.

---

> ### Author Response · Authors · 2025-03-07
> **Response to reviewer JgxJ (2/2)**
>
> > I recommend the authors to investigate how WANN works for a simple toy model.
>
> We thank the reviewer for suggesting this interesting toy experiment, which we now included in Appendix B.2. Specifically, we generated $1,000$ samples from two Gaussians and injected $40\%$ symmetric noise. For clarity, we kindly suggest to read the description of the experiment and plot.
>
> From the bottom-right plot in the additional Figure B.1 (Appendix B.2), it is possible to observe that clean samples (blue) tend to have higher reliability scores, whereas noisy samples (red) cluster at lower or intermediate values. Points near decision boundaries or mislabeled neighbors also receive moderate or low reliability scores. This observation supports our intuition that WANN utilizes local consistency in the embedding space to effectively distinguish clean from noisy samples.
>
> > I think that WANN can be applied to tabular data without modification. Is there any reason to focus only on classification of embedding vectors?
>
> We thank the reviewer for this valuable perspective on extending WANN to tabular data. In principle, WANN can indeed be applied to any domain, tabular or otherwise, as long as we can obtain a meaningful semantic space. While our current work focuses on image embeddings derived from large vision foundation models, the same concept extends to language, audio, or multimodal embeddings, assuming they capture the semantic relationships necessary for defining reliable neighborhoods. In practice, tabular data often lacks the kind of rich semantic structure found in image or language embeddings, so additional feature engineering or domain-specific embedding techniques might be required. Nonetheless, WANN's underlying idea, i.e., using local consistency in a robust feature space, remains applicable, and exploring its effectiveness on tabular datasets is a promising direction. We have now included this mention in the discussion section.

---

### Comment · Reviewer_JgxJ · 2025-02-23
**Questions/comments**

1. What happens when there is no $k$ satisfying the Eq (1)? I think that this situation could occur frequently for
noisy-labeled data in the minor class.  I


2. In the last paragraph of page 4, $k$ would be $\mathcal{H}(x,y,\mathcal{X})$ (or $eta$). $k$ is an argument used to define $\mathcal{H}(x,y,\mathcal{X})$.

3. I understand the idea of using the smallest number of nearest neighbors for correct classification (i.e. k) for measuring the reliability
but I am not sure that 1/k is an optimal choice. For me, 1/k seems to be too naive.

4. I understand the intuition of (4) , but it seems too naive. Are there any theoretical justifications for (4)?
I worry that (4) depends only on the nearest datum but not on multiple nearest neighbors.

5. As far as I understand, 'Filtered dimensionality reduction' is needed since nearest neighbor methods are susceptible to the curse of dimensionality. I wonder how sensitive the nearest neighbor method to the dimension of data when obtaining the reliability scores. I think
that nearest neighbor is still sensitive and thus the reliability scores may not be informative when the dimension of data is high.

6. In numerical studies, it seems that WANN outperforms other methods using specially designed loss functions for noisy-labels.
However, I could not understand intuitively why WANN is better? More detailed explanations would be necessary.

7. Is it possible to combine the reliability score and specially designed loss function? For example, we could learn a prediction model
by minimizing the weighted empirical average of the specially designed loss functions where the weights are proportional to the proposed reliability scores. This idea would be promising since the nearest neighbor method is difficult to be deployed.

8. To justify the proposed method (e.g. explain why WANN is better than existing loss-based methods), I recommend
the authors to investigate how WANN works for a simple toy model such as two Gaussian mixture model.

9. I think that WANN can be applied to tabular data without modification. Is there any reason to focus only on classification of embedding vectors?

---

### Author Response · Authors · 2025-03-07
**General statement**

We sincerely thank the reviewers for their time and effort in reviewing our work, as well as for their positive feedback and constructive suggestions. We are grateful for the recognition that our method achieves comparable or superior performance to the state of the art while being straightforward and intuitive (y5QB, JgxJ), and computationally efficient (y5QB). Furthermore, we appreciate the recognition of our novel approach to address the label noise problem at inference time (XUqU), along with the acknowledgment of its well-motivated, technically sound nature and the extensive experimental evidence supporting its benefits (XUqU, y5QB). Lastly, we are pleased that the paper is regarded as well-written and easy to follow (y5QB). We address reviewers' concerns below and welcome additional queries the reviewers may have.

---

### Decision · Action_Editor_f1Gf · 2025-03-23

**Recommendation:** Accept with minor revision

**Comment:**

The paper proposes a method for addressing the label noise problem. Interestingly, they address this problem at the inference stage (rather than during training), by leveraging pretrained foundational models and a weighted nearest-neighbour method.

Reviewers appreciated the experimental results, and the fact that the method is relatively simple to apply, meaning that it can be readily used in practice.

And whilst reviewers appreciated the overall idea of addressing the label noise problem at inferene-time which is quite novel, they also felt that the idea was quite simple and lacked stronger theoretical motivations.

On the balance, the decision is to accept the paper as its strengths outweigh its weaknesses. However, the paper is not being recommended for the ICLR Journal-to-Conference track, or any certifications.

**Audience:**

All reviewers agreed that some of TMLR's audience would find the paper relevant.

**Claims And Evidence:**

All reviewers felt that the claims in the submission are substantiated with experiments.

---

> ### Author Response · Authors · 2025-03-31
> **Response to Action Editor**
>
> The authors sincerely thank the Action Editor and reviewers for their time and thoughtful feedback, which helped to improve the quality of our work. We have submitted the camera-ready version, incorporating the suggestions provided during the review process.